# A New Generation of Minor-Groove-Binding—Heterocyclic Diamidines That Recognize G·C Base Pairs in an AT Sequence Context

**DOI:** 10.3390/molecules24050946

**Published:** 2019-03-07

**Authors:** Ananya Paul, Pu Guo, David W. Boykin, W. David Wilson

**Affiliations:** Department of Chemistry and Center for Diagnostics and Therapeutics, Georgia State University, 50 Decatur St SE, Atlanta, GA 30303, USA; apaul@gsu.edu (A.P.); pguo@gsu.edu (P.G.); dboykin@gsu.edu (D.W.B.)

**Keywords:** DNA G·C base pair recognition, mixed base pair DNA sequences, sequence selectivity, heterocyclic diamidine, σ-hole, thiophene-*N*-methylbenzimidazole, benzimidazole, aza-benzimidazole, biosensor

## Abstract

We review the preparation of new compounds with good solution and cell uptake properties that can selectively recognize mixed A·T and G·C bp sequences of DNA. Our underlying aim is to show that these new compounds provide important new biotechnology reagents as well as a new class of therapeutic candidates with better properties and development potential than other currently available agents. In this review, entirely different ways to recognize mixed sequences of DNA by modifying AT selective heterocyclic cations are described. To selectively recognize a G·C base pair an H-bond acceptor must be incorporated with AT recognizing groups as with netropsin. We have used pyridine, azabenzimidazole and thiophene-*N*-methylbenzimidazole GC recognition units in modules crafted with both rational design and empirical optimization. These modules can selectively and strongly recognize a single G·C base pair in an AT sequence context. In some cases, a relatively simple change in substituents can convert a heterocyclic module from AT to GC recognition selectivity. Synthesis and DNA interaction results for initial example lead modules are described for single G·C base pair recognition compounds. The review concludes with a description of the initial efforts to prepare larger compounds to recognize sequences of DNA with more than one G·C base pairs. The challenges and initial successes are described along with future directions.

## 1. Introduction

### Overview of DNA and RNA Targeting Challenges

The interaction of designed, synthetic, small organic molecules with macromolecules is a fascinating and important current research area for both biophysical studies and for the design of new reagents for selective binding applications as well as for therapeutic development. Compounds that bind to proteins have become important molecular probes but also account for many of our most active current drugs [1,2,3]. The design of agents that selectively target sequences or structures of DNA or RNA has progressed more slowly due to their more repetitive structures and properties [4,5,6,7,8]. The nucleic acids have only four basic components and a high negative charge compared to the complexity of components and charged groups in proteins. The generally repetitive duplex structure of DNA as well as duplex sections of RNA makes selective targeting more of a challenge. Compounds which bind with some selectively to nucleic acids have, however, produced important cellular probes, chromosomal dyes, and a number of important therapeutic agents [9,10,11,12,13,14]. Among the earliest discovered agents that interact with nucleic acids are compounds that selectively bind to A·T base pair (bp) sequences in the DNA minor groove [15,16,17,18,19,20]. These agents generally have some flexibility in AT sequence binding but strongly exclude G·C bps. In duplex regions G·C bps have a third hydrogen bond, compared to A·T bps and the extra group protrudes into the minor groove. The extra group sterically hinders most of the original minor groove agents from sliding deeply into the groove to interact with the edges of bps in the groove [15,16,17,18,19,20,21]. The G·C bps are also more electropositive than A·T bps and this also hinders binding of cationic minor groove agents [22,23] such as those shown in Figure 1, for example. Compounds that bind in the nucleic acid grooves have been especially attractive for therapeutic development since they are generally less mutagenic and less toxic than intercalators [24,25].

As our knowledge of RNA structures and functional complexity has expanded to RNAs with important cellular functions such as control of gene expression and translation, the number of research groups involved with design and testing of agents for selectively binding to specific RNA base pair groups and structures has rapidly increased [26,27,28,29,30,31,32,33]. Because of the structural complexity of RNA, relative to DNA, designing agents to bind to RNA is quite a different and difficult challenge compared with DNA or protein targeting. To date there are few designed agents that selectively target RNA but it is an area of great potential.

## 2. AT Specific Minor Groove Drugs: Antiparasitic Agents

AT-specific heterocyclic cations such as Hoechst 33258 and DAPI have been successful in staining components in cells, particularly chromatin. Other AT specific cations such as the diamidines furamidine, pentamidine, berenil and related diamidines (Figure 1) have had significant success as antiparasitic agents and pentamidine has been used in humans for over 50 years [25,34,35,36,37,38,39]. Interestingly, however, no systematic effort to redesign these agents into mixed base pairs recognition compounds has been described previously. Mascareñas and coworkers have made several novel and important designed bis-benzamidine compounds such as BAPPA (Figure 1) [40] which can recognize pure AT bps with very high affinity. BAPPA and other pentamidine analogues, where the pentamidine oxygens are replaced by NH, have much improved spectroscopic and solubility properties [12,41]. One synthetic bis-benzamidine has a central pyridine group that gave enhanced GC bp recognition in an AT sequence context [42]. New ideas to redesign successful A·T bp specific diamidines, so that they have significantly broadened sequence recognition capability and potentially expanded uses, are reported in this review.

Starting in the mid-1980s, design of minor groove agents to recognize GC bps took a step forward with ideas from the Dickerson and Lown laboratories for modifications of the minor groove binding polyamides netropsin (Nt) and distamycin (Dst), which specifically recognizing A·T bp. The idea was to make new synthetic compounds that could also bind to a G·C bp [43,44]. They proposed replacing one or more pyrrole groups in Nt and Dst with an imidazole group so that the extra N in imidazole could form an H-bond with the G-NH_2_ in the minor groove of a G·C bp [43,44,45]. The idea attracted attention in several additional laboratories and significant success in GC recognition was achieved by the Dervan laboratory by connecting two recognition units to make a hairpin polyamide [46,47,48]. The synthetic polyamides are very attractive reagents, but solution difficulties, aggregation, and poor cell uptake have limited their applications [49,50,51]. Several collaborating laboratories at the University of Strathclyde have designed, synthesized and tested polyamides with successfully modified heterocycles to improve their solution properties, DNA recognition and therapeutic potential [52].

Our interest in nucleic acid binding compounds began with a desire to design new types of agents to treat parasitic diseases [36,53]. For most parasitic diseases there are limited small molecule therapeutic options, and there are few or no satisfactory vaccines. Unfortunately, drug design for many parasitic diseases has received little recent attention and funding, since these are largely diseases of economically disadvantaged countries. Trypanosome parasites, for example, cause diseases like sleeping sickness and Chagas diseases, particularly in Africa, Central, and South America [54,55,56,57]. There are very few effective drugs to treat these deadly diseases. Many of the available compounds that are used have been available for many years and have significant toxicity [58].

As described above, most classical minor groove binding agents have AT interaction specificity. The malaria genome, for example, is quite AT-rich and many AT specific DNA binding compounds have excellent activity against malaria [59,60,61]. The mitochondrial genome of trypanosomes that cause human African sleeping sickness is also very AT-rich and presents an excellent AT selective DNA binding site array [62]. The fact that pentamidine (Figure 1), an AT specific DNA minor groove binder, is an active agent for sleeping sickness has provided a valuable lead into this system [63]. Our collaborative synthetic and biophysical groups have prepared and tested a wide variety of heterocyclic amidine derivatives, and we have found a number of compounds that are highly active against sleeping sickness [25,36,53,64,65]. Development of these compounds produced a prodrug of furamidine (DB75 in Figure 1) that has moved to Phase III human clinical trials [25,66]. Because these heterocyclic diamidine compounds are intrinsically fluorescent, they can be directly observed in cells [64]. Fluorescence microscopy showed that the compounds selectively target the AT-rich mitochondrial kinetoplast DNA [64,67,68]. After addition of the compounds to a trypanosome infected mouse, the kinetoplast became highly fluorescent, then disintegrated followed by disappearances and death of the trypanosomes. These initial, relatively simple compounds provided a proof of concept that diamidines could be used effectively in humans. 

Over a century ago Paul Ehrlich raised the question “ --- do the trypanosomes possess in their cells definite groupings which govern the captivation of definite chemical substances?” Cellular fluorescence microscopy of trypanosomes treated with furamidine and related compounds provides a definite “*Yes*” answer to his question.

## 3. Design of Compounds that Expand Past Pure AT Sequence

We are also continuing with the design of new compounds that will be more specific in targeting kinetoplast sequences with greater affinity, selectivity, and potentially reduced toxicity. The kinetoplast genome contains thousands of small circular AT-rich DNAs that code for guide RNAs that control mRNA editing in trypanosomes. Since the small DNAs are AT-rich, they are ideal targets for heterocyclic diamidines [36,64,68]. The AT sequences in the kinetoplast typically have many single G·C bps flanked on both sides by AT sequences. We reasoned that compounds that could target the single G·C bp, as well as the flaking AT sequences, would have much greater selectivity in binding to the kinetoplast with reduced toxicity.

Agents that can recognize mixed base pair sequences of DNA containing combinations of A·T and G·C bps are an important development area for minor groove binders. As described above, the development took a major step forward with modification of the minor groove binders, netropsin (Nt) and distamycin (Dst) [43,44,45]. The proposal was to replace one or more pyrrole groups in Nt. and Dst. with *N*-methylimidazole groups [43,44,45]. As has been described, the extra N in *N*-methylimidazole relative to pyrrole should allow that group to H-bond with the G-NH_2_ that protrudes into the minor groove. Success in mixed sequence binding with both A·T and G·C bp recognition with polyamides required the finding by Wemmer and coworkers that it took two stacked or linked recognition units to make an effective minor groove polyamide for GC targeting [69,70]. Although several groups have worked with synthetic polyamides, solution difficulties have limited their applications [49,50,51]. Polyamide work on G·C bp targeting has continued to focus on replacement of pyrrole by *N*-methylimidazole or other groups that would give an H-bond acceptor that could bind to the G-NH_2_ group that projects into the minor groove [71,72,73,74]. Research in several laboratories has expanded past *N*-methylimidazole to new GC recognition agents with the goal of preparing polyamides with broader therapeutic potential [75,76,77].

Our goal was to prepare entirely new types of non-amide compounds that could bind to mixed A·T and G·C bps containing sequences but would also have varied chemical and structural properties that would allow them to effectively target a variety of different cell types (Figure 2). The initial research in the area has produced quite different compounds that are also quite different from the polyamides and are without pyrrole and imidazole groups [78,79,80]. Our three most successful compounds in targeting a single G·C bp in an AT sequence are shown in Figure 2 [78,79,80]. All three of these compounds are undergoing additional development to increase affinity and selectivity for the target sequence type with a single G·C flanked by A·T bps. Brief synthetic schemes for the compounds of Figure 2 that recognize a single G·C bp in an AT sequence are given below.

### 3.1. Synthetic Procedures for DB2429, DB2457 and Analogues

The syntheses of the N-substituted benzimidazole heterocycles are outlined in Scheme 1 [80,81]. In this series the H-bond acceptor unit is the σ-hole array 1,4-N--S or 1,4-N--Se provided by N-substituted benzimidazoles and either a thiophene or a selenophene. Reaction of the readily available bromoheteroaryl aldehydes **1**, when X = S or Se provides a key component for σ-hole formation, with 4-cyanophenylboronic acid under standard Suzuki coupling conditions conveniently provides the 5-(4-cyanophenyl)-2-formyl 5-ring heterocycles **2**, which can be used to prepare both the tri- and tetra-heteroaryl target molecules **4** and **10**. Condensation, cyclization, and oxidation, mediated by sodium metabisulfite, of the aldehydes **2** with 3-amino-4-(alkyl or aryl amino) benzonitrile, which provides the second component needed for σ-hole formation, gives the bis-nitriles **3**. These can be readily converted into the desired diamidines **4** using LiN(TMS)_2_ in THF. In a similar sequence, the aldehydes **2** are allowed to react with the arylphenylenediamines **8**, again mediated by sodium metabisulfite, to produce the bis-nitriles **9**.

These are then converted to the diamidines **10** as previously described. In order to obtain the arylphenylenediamines **8**, nucleophilic aromatic substitution reactions between 4-bromo-1-fluoro-2-nitrobenzene (**5**) and various aliphatic and aromatic amines are employed which yields the 4-bromo-*N*-alkyl and *N*-aryl-2-nitrobenzenes **6**. The nucleophilic aromatic substitution reaction with aliphatic amines was achieved at room temperature in ethanol, whereas, the less basic aryl amines require heating in dimethylacetamide in the presence of Cs_2_CO_3_ at 160 °C. Suzuki−Miyaura coupling of substituted 4-cyanophenylboronic acids with the various 4-bromo-*N*-alkyl and *N*-aryl-2-nitrobenzenes under standard conditions provides the nitro biphenyl analogs **7**. The arylphenylenediamines **8** required for formation of the *N*-substituted benzimidazoles are obtained by stannous chloride reduction of the nitro groups of **7** and are subsequently allowed to react directly without characterization. Including the tri- and tetra-heteroaryl target molecules available for single G·C bp binding studies, we have designed and synthesized almost 70 target compounds containing σ-hole modules.

### 3.2. Synthetic Procedures for DB2120 and Its Analogues

In the Scheme 2 examples a library of new molecules is prepared by including the H-bond accepting group in the linker connecting two terminal benzimidazole-diamidine units **12** in Scheme 2 [78]. The latter are well studied in the field of AT sequence-specific DNA minor groove binders which have been shown to exhibit excellent nuclear uptake and transcription factor inhibitory activity [82]. In this series we have altered the linker to provide H-bond accepting units capable of G·C bp recognition but to retain some conformational flexibility. The preparation of the symmetrical dialdehydes **11** needed to construct the terminal benzimidazole-diamidine units, employs coupling *p*-hydroxybenzaldehyde with the various di-halo substituted linkers in the presence of K_2_CO_3_/Cs_2_CO_3_ in dimethylformamide. The compounds **11** are allowed to react with 4-amidino-1,2-phenylene-diamine hydrochloride hydrate in anhydrous ethanol, mediated by 1,4-benzoquinone, to yield DB2120 and various analogues.

### 3.3. Synthetic Procedure for DB2277

In this series, an azabenzimidazole is employed as the H-bond accepting group, an essential unit for recognition of a G·C bp [79]. The synthesis of the most selective GC binding compound of this type, DB2277, an azabenzimidazole with a flexible -O-CH_2_ linker, begins with coupling of commercially available 2-amino-6-chloro-3-nitropyridine and 4-hydroxymethylbenzonitrile in the presence of sodium hydride in dimethylformamide (Scheme 3). The intermediate nitro compound is reduced by iron dust in a 2-propanol/water mixture at reflux which provides the diamine compound **13**. In the following step, 4-cyanobenzaldehyde is allowed to react with diamine **13** in the presence of sodium bisulfite in DMF at reflux to provide the *bis*-nitrile substituted azabenzimidazole intermediate **14**. The desired hydrochloride salt of the diamidine DB2277 is obtained by stirring the intermediate **14** in ethanolic HCl, applying Pinner methodology where the bis-imidate ester hydrochloride is ultimately converted to the hydrochloride salt of the diamidine in the presence of ethanol saturated with ammonia gas [71].

## 4. Increasing the Design Diversity of Compounds to Specifically Recognize the Minor Groove

Until recently recognizing mixed base pair sequence in the minor groove of DNA has largely continued to follow the original ideas with pyrrole and imidazole and related groups with amide linkers in polyamides. Such compounds have an advantage of design simplicity, but have been plagued by solution difficulties and poor cell uptake [49,50,51,83]. The compounds in Figure 2 have good solution properties, and cell uptake and they were designed with a structural and chemical diversity that gives them an advantage in targeting different cell types [82]. The three different GC recognition groups, pyridine [78,84], azabenzimidazole [79,85,86] thiophene-*N*-methylbenzimidazole (-*N*-MeBI) [80,81], in specific minor groove recognition modules offer a much-needed breakthrough in mixed sequence binding with a range of solution properties. Each of these compound types represents the results of extensive development work to enhance sequence selectivity and affinity. The pyridine compounds were found to require the adjacent –CH_2_-O- linker to give acceptable affinity and selectivity [78]. In a similar manner, the azabenzimidazole (DB2277) was found to require the flexible linker –O-CH_2_- to function well [86]. Compounds without that group bind with much lower selectivity to target AT sequences with a single G·C bp. The same extensive development yielded the initial thiophene-*N*-MeBI compound in that series (Figure 2 and Figure 3, DB2429). We cannot present further development of all three compounds in the space available and will, therefore, focus on the thiophene-*N*-MeBI module in the remainder of this review as an example for all three compounds. 

### 4.1. Initial Concept and Development

The initial concept for development of the thiophene-*N*-MeBI module was that the thiophene S and *N*-alkyl benzimidazole unprotonated *N* could form a σ-hole based interaction that would preorganize the module appropriately for recognization of the shape of the minor groove [80]. This shape also points the unprotonated *N* of *N*-alkylBI into the minor groove to H-bond with the G-NH_2_ group in that groove. The σ-hole bonding is a highly directional noncovalent interaction between the covalently bonded *S* atom in thiophene and the lone pair on the nucleophilic *N* in BI. When one of the half-filled p orbitals in the thiophene S atom is involved in a covalent bond, results in electron deficiency in the outer (noninvolved) lobe of the orbital that is a σ-hole. If the remainder of the molecule is sufficiently electron-withdrawing relative to the polarizable *S* atom, then there is a positive electrostatic potential associated with the σ-hole that is responsible for the outer-lobe electron deficiency. This positive potential can interact attractively with the negative lone pair on the BI-N; The resulting noncovalent interaction, directed roughly along the extension of the C-S covalent bond in thiophene, is a σ-hole bond. The *S* in thiophene can have two positive σ-holes, approximately along the extensions of the C-S bonds but there is only one interaction in DB2429 with the one *N*-MeBI. Thiophene C-S single bonds in general provide a relatively positive electrostatic potential that can form an interaction with electron donating atoms such as the unsubstituted *N* in N-MeBI, a 1,4 N···S interaction (Figure 2 and Figure 3) [87].

The thiophene-*N*-methylbenzimidazole (*N*-MeBI), DB2429 in Figure 2 and Figure 3, can specifically bind to a mixed sequence DNA with a single G·C bp with flanking A·T bps and it was the first successful compound in the development of the thiophene-*N*-MeBI series [80]. Expanding DB2429 with another phenyl to better cover the A·T bps that flank the central G·C bp gave DB2457 (Figure 3). These compounds bind strongly with selectivity for the target single G·C bp sequence and are a significant step forward in our molecular design and synthesis project for recognition of mixed bp DNA sequences [78,79,84]. Both the *N*-MeBI and thiophene are necessary for strong, specific interactions [80]. If a furan or pyridine, for example, replaces the thiophene the binding strength and specificity are both reduced [80]. There is a remarkable reversal of binding specificity when the *N*-Me group is replaced by an NH group and a purely AT selective compound is obtained. Compounds with the *N*-Me substitution replaced by other similar groups also have reduced affinity and selectivity [80]. GC binding by compounds with the thiophene-*N*-MeBI is an important new observation in ideas for the design of agents to recognize mixed bp DNA sequences.

The selectivity for binding to the single G·C bp sequence instead of a pure AT sequence is only about a factor of 10 for DB2429 while it is increased to 50 for DB2457 [80]. One idea for additional increases in the selectivity for single G·C bp sequences is to take advantage of DNA microstructural variations, such as the wider minor groove in G·C bp containing regions and the difference in minor groove width (MGW) for the single G·C bp sequence versus pure AT sequences. We have used the DNA Shape algorithm of Rohs and coworkers to evaluate the minor groove widths [88,89,90,91]. In Figure 4 the MGWs for closely related target sequences AAATTT, AAAGTTT and AAAGCTTT in a hairpin duplex context are compared [88]. The MGW is different for all three in the central target region, smaller for AAATTT, larger for the single G·C bp sequence and much larger for the two G·C bps sequence. An idea for new agents is that compounds with increased bulk/steric crowding might favor the wider, mixed bp sequences over pure AT. To test this idea the following two general changes in the compound structure and functional groups were made with the goal of increasing specificity: (i) increasing compound bulk primarily by replacing the *N*-Me group of *N*-MeBI with groups of increased size and (ii) modifying the overall twist in the compound linked aromatic core structure by adding appropriate substituents (Figure 3).

As we move to prepare compounds with therapeutic potential by inhibiting DNA-protein complexes, such as specific transcription factors, two things are very important: the selectivity ratio and the ability of the compounds to enter different cell types. At least some substituent changes at the *N*-BI position and aromatic core should have positive effects on both of these critical factors. We were able to successfully synthesize compounds with a variety of substituents at the *N*-BI position and other compounds with substituent modifications on the aromatic groups of the thiophene-*N*-substitution BI module [81]. Representative examples with the modified compounds along with results for their DNA interactions are given here.

### 4.2. Biosensor-SPR Binding Affinity Results for Modified Thiophene-N-RBI Modules

In Table 1, results for seven derivatives of the extended compound, DB2457 are shown. Converting the *N*-Me to *N*-Et gives similar binding to the single G·C bp target sequence with improved selectivity over the pure AT and the GC sequence (not shown). The *N*-*i*-Pr compound, DB2708, is even more improved with a selectivity ratio of 255 for AT and 56 for GC. Enlarging the substituent to isobutyl, DB2711, yields a new potential therapeutic candidate with slightly reduced binding to the single G·C bp sequence, but remarkably no detectible binding to either the pure AT or the GC sequence (Table 1).

SPR sensorgrams for DB2711 with the three DNAs are shown in Figure 5 to illustrate how NB (not binding) is defined experimentally. Results in all three sensorgrams in Figure 5, were collected up to the same maximum concentration. As can be seen under these conditions, neither the sensorgrams for the AAATTT or the two G·C bps sequence moved significantly off of the baseline at the maximum concentration and this is our operational definition of “NB”. The sensorgram results for the single G·C sequence show quite significant binding with slow dissociation kinetics. The black lines in the figure are for a single site kinetics fit to the SPR results. The *K*_D_ values are given in Table 1 and the kinetics values are discussed below.

Cyclic *N*-substituents were also explored for binding affinity and specificity. The cyclopentyl derivative, DB2714, has good affinity for the single G·C bp and high selectivity but is not quite as good as the isobutyl derivative, DB2711. The cyclohexyl derivative, DB2727, has weaker binding to the single G·C bp sequence and poor selectivity with GC. Clearly, the more bulky cyclohexyl has stronger binding to the wider GC minor groove than the cyclopentyl compound but does not fit as well in the AAAGTTT sequence.

To explore *N*-aromatic groups, an *N*-phenyl derivative, DB2740, was prepared. It was found to have good binding to the single GC sequence, no detectible binding to AT but poor selectivity for GC. These results suggest that the twist of the phenyl substituent gives a good fit to the target single GC sequence but poor fit to the narrow minor groove of pure AT. The binding to GC is better, probably because the twist of the phenyl with the lowest energy gives a good fit to the wider GC. A number of other derivatives have been prepared but with no improvement over the isobutyl derivative, DB2711 [81]. 

In order to change the overall shape of the core aromatic-amidine system, several derivatives were prepared. DB2762 with a -CF_3_ group *ortho* to an amidine gave no detectible binding to AT and GC but relatively weak binding to the single GC sequence. Clearly the large twist of the phenyl-amidine system caused by the –CF_3_ group does not fit any of the sequence-dependent minor groove shapes. DB2759 with an *ortho* –Cl substituent is a much improved compound. It has strong binding to the single GC but no detectible binding to AT and GC. DB2759 joins DB2711, the *N*-isobutyl compound, as our two best therapeutic leads. Other derivatives were not as good as these compounds and are not shown.

In order to understand the difference among the –H, -Cl, and –CF_3_ substituents at the same phenyl-amidine substitution position, torsional angle maps were constructed in the *SPARTAN* 16 software (Figure 6) [92]. The lowest relative energy represents the most stable twist structure and also shows the energy needed to change the shape into a more planar conformation to fit the DNA minor groove. Around 10 kJ/mol is needed to twist the 30°structure of –H ortho to a planar conformation of the phenyl-amidine group. For –Cl, 12 kJ/mol is needed to give a similar result. The -CF_3_ modification with its larger bulk, however, needs more energy to twist its larger torsional angle into a planar conformation. All of the above results show that, when lower energy is needed to reach a shape that can bind to the minor groove, stronger binding to the target sequence is obtained. Different final twist angles than 0°can be used but the general conclusions are the same.

### 4.3. Biosensor-SPR Binding Kinetics Results for Modified Thiophene-N-RBI Modules

Five compounds from Table 1 with a variety of *N*-substituents were used to compare the kinetics of association, *k*_a_, and dissociation, *k*_d_, for the AAAGTTT sequence. The binding association and dissociation kinetics were quite rapid with the AAATTT and AAAGCTTT sequences and could not be determined by SPR methods. For comparison of the kinetics results for the five test compounds with AAAGTTT, log *k*_a_ and log *k*_d_ values are plotted in Figure 7 with results in Table 1. The blue dashed lines in Figure 7 compare compound binding affinities on a logarithmic scale. The *N*-Me reference, DB2457, and strong binding compounds, DB2740 and DB2708, are located on a *K*_D_ line with a slope of 3~4 nM. The other two compounds, which bind somewhat more weakly, are located near the 10 nM *K*_D_ blue dashed line in Figure 7. The on-rates and off-rates for the compounds change in a correlated manner to give similar *K*_D_ values.

For the *N*-substituted compounds in Figure 7, the *K*_D_ values with AAAGTTT cover a narrow range, from around 4 to 13 nM. As can be seen in Figure 7, the kinetics results for these compounds also cover a relatively narrow range. The *N*-Me (DB2457) and *N*-Ph (DB2740) compounds have the lowest *K*_D_ values in this set and this arises from their similar *k*_d_ values. DB2711 with an isobutyl substituent has the fastest disassociation rates with the weakest binding (9.6 nM) to AAAGTTT among the compounds in Figure 7. These results suggest that the very dynamic structure of the butyl substituent interferes with binding in the minor groove giving compounds fast association and dissociation with the highest *K*_D_ values in this set. These observations indicate that the faster association results from a lack of optimum penetration into the minor groove and, as a result, the compounds also dissociate more rapidly. It should be noted, however, that even the weakest binding of the compounds in Figure 7 bind very strongly, in the 10 nM *K*_D_ range. It should also be noted that the relatively small variation in *K*_D_ for the compounds in this set arises primarily from small variations in *k*_d_ values (Figure 7). The *k*_a_ values have even less variation and are quite large.

### 4.4. A DB2708 Model from Molecular Dynamics Simulations

In order to get a better understanding of how the larger substituents on the thiophene-*N*-RBI fit into the minor groove, a molecular dynamics (MD) simulation was conducted on the DB2708 *N*-isopropyl compound DNA complex by using methods that we have previously described [93,94]. DB2708 was docked into our DNA sequence with a single G·C bp and flanking A·T bps. DB2708 force constants for the MD simulation were determined as previously described [94,95]. The MD simulation was conducted for 500 ns, and describe structure for the complex of DB2708 in the AAAGTTT binding site is shown in Figure 8. The MD results were evaluated in detail to determine what major features of the DB2708-DNA complex are responsible for the excellent stability and specificity ratios of the *N*-modified compounds.

There are three critical H-bonds in the complex, two from amidine -NHs to T=O groups that are an average of 2.8 Å in length. The third strong H-bond with 2.2 Å bond length is from the G·C bp G-NH in the minor groove to the unsubstituted imidazole N in the BI group of DB2708 and accounts for a significant amount of the binding selectivity for DB2708 and other thiophene-*N*-RBI compounds length. The complex is additionally stabilized by water molecules that link DB2708 to the DNA binding site. The amidines, for example, form numerous dynamic H-bonds to water molecules that move in and out of the minor groove. Additional H-bonds with DB2708 are frequently formed by water molecules with A·T bp groups at the floor of the minor groove, and these interactions help link the compound to the specific binding site in the groove and stabilize the complex. Selectivity in binding is also enhanced by the −CH group of the six-member ring of BI that points into the minor groove. This −CH forms a dynamic close interaction with the −C=O of the dC base of the central G·C bp as well as to a dT=O on an adjacent A·T bp (Figure 8). This interaction enhances the binding selectivity as well as the binding affinity.

As seen with other minor groove binders, additional stabilizing interactions are formed by phenyl −CH groups of DB2708 that point to the floor of the minor groove. These −CH groups form significant dynamic interactions with dA-N3 groups on the bases at the floor of the groove (see Figure 8 for an example). DB2708 tracks well along the minor groove with a twist and shape to match the minor groove curvature. As a result of this shape match, the aromatic-system of DB2708 forms strong interactions with the sugar-phosphate walls of the minor groove. The cationic amidines of DB2708 also form electrostatic interactions with the backbone phosphates that result in Na^+^ release and an entropy increase that is a stabilizing component, for complex formation.

The release of strongly bound water molecules from the minor groove as DB2708 binds also provide a favorable entropy term to the binding energetics. The sum of these numerous weak, stabilizing interactions results in the observed strong binding of DB2708, a ΔG of binding of 11.5 kcal/mol. The isopropyl group of DB2708 forms a tight interaction with the DNA backbone in the G·C bp region and this is not possible with the narrower minor groove of a pure AT sequence (Figure 4). The *K*_D_ for binding to the AAATTT sequence, for example, jumps from 200 nM with DB2457, the *N*-Me compound, to around 1000 nM with DB2708 due to the *N*-isopropyl group (Table 1) and this provides a significant increase in selectivity. It seems that many of the *N*-RBI derivatives can match the minor groove microstructure in AAAGTTT better than with AAATTT.

## 5. Extending the Capability to Recognize Mixed BP Sequences of DNA to More Than a Single G·C BP

While the three types of compounds described above for converting pure AT DNA minor groove compounds to recognize mixed DNA sequences with a G·C bp flanked by A·T bps was a major step forward, it is clearly only the first step. To make this discovery widely useful, recognition of different DNA sequences, such as, for example, TF promoters, with several G·C bps is essential [96,97,98,99]. An attractive way to start on the extended sequence compounds is by combining modules from the three single G·C bp recognition compounds. In this section we will describe our initial steps in this direction with the thiophene-*N*-MeBI module.

The critical question is how to combine single GC recognition modules, such as the thiophene-*N*-MeBI in DB2429, to selectively recognize longer, more complex DNA sequences with additional G·C bp. Although progress in targeting DNA with proteins has been excellent, there is limited progress in the variety of compounds designed to target mixed base pair (bp) DNA sequences. Compounds are needed that can selectively recognize two (or more) G·C bps in DNA sequences such as (A/T)_x_-(G/C)-(A/T)_n_-(G/C)-(A/T)_y_ where “n” can be 0–5 bps (Figure 9B) and x and y are variable. 

Because of the considerable difficulty of preparing longer, complex compounds that have the correct curvature for the minor groove and proper indexing of H-bonding groups to match the array of DNA H-bonding groups at the floor of the minor groove, we started with relatively simple, flexible combinations. For the initial test of this approach, a flexible linker, –O-(CH_2_)_3_-O-, was used to connect GC recognition molecular modules based on DB2429 (Figure 9A). The design strategy provides flexibility to match the shape and curvature of the DNA minor groove with appropriate length so that modules are spaced to H-bond and interact strongly with a GAAAC DNA sequence. Such modules are reasonable to synthesize and can recognize a full turn or more of the DNA double helix.

### Synthetic Procedures for DB2528 and DB2604

The syntheses of the two GC recognition compounds are outlined in Scheme 4. Protected 5-(4-hydroxyphenyl)thiophene-2-carboxaldehyde (**15**) is readily prepared through a standard Suzuki coupling reaction between the acetal of 5-bromo-2-thiophene carboxaldehdye and 4-hydroxy-phenylboronic acid. Standard Williamson ether synthesis methodology was employed to link 1,3-dibromopropane with 5-(4-hydroxyphenyl) thiophene-2-carboxaldehyde to produce the bisaldehyde **16**. The target products bis-amidino-*N*-methylbenzimidazolethiophenes are prepared from amidino-*N*-methylphenylenediamines by coupling with the symmetrical bis-thiophenecarboxaldehyde in the presence of benzoquinone (Scheme 4) [100].

DNA thermal melting experiments (*T*_m_) offer a rapid, screening method for compounds binding to different DNA sequences [101,102]. Binding to DNA gives an increase in *T*_m_ that is related to binding affinity. Hairpin DNA oligomers, which have monophasic melting curves, were used in the assay. DB2528 and analogs (Figure 9A) are the first linked compounds to test this concept for two G·C bps binding. DB2528 does not bind well to the single GC test sequence (AAAAGTTTT, Δ*T*_m_ = 2 °C, but in an exciting result it binds with a Δ*T*_m_ of 9 °C with a target two G·C bps sequence, GAAAC. As desired, it has lower *T*_m_ values with longer, GAAAAC and GAAAAAC sequences (6 °C) and shows no significant binding with an all AT sequence or with related GC sequences (GC, GAC, GAAC), all with Δ*T*_m_ < 2 °C. As with the simpler compounds above, replacement of *N*-MeBI in DB2528 with benzimidazole (BI) gives a minor groove binder that binds very strongly to extended AT sequences with weaker binding to G sequences (not shown). Circular dichroism (CD) studies with GAAAC and DB2528 show a strong positive induced CD peak at the compound absorption maximum that indicates minor groove binding. There is little CD change on addition of DB2528 to AT sequences in agreement with weak binding [91].

As above, the interaction affinities of DB2528 were quantitatively evaluated by biosensor-SPR methods with the mixed base pair sequences. DB2528 binds strongly with GAAAC, in agreement with *T*_m_ results, and global kinetics fitting yielded a single binding site with a *K*_D_ of 5 nm, a rapid on-rate (*k*_a_ = 2.8 ± 0.5 × 10^6^ M^−1^s^−1^) and a slow off-rate (*k*_d_ = 1.7 ± 0.4 × 10^−2^ s^−1^) (Figure 7). The dissociation kinetics of DB2528 with the GAAG sequence are much faster than with GAAAC and the *K*_D_ = 149 nM with GAAG. Pure AT and single G sequences were also studied by SPR, and DB2528 has no detectable binding with either of them. These results indicate excellent selectivity and strong binding for a two G sequence that has a specific distance between the two G·C bps. In order to broaden the solution properties and enhance potential cell uptake in future therapeutic applications, we also synthesized DB2604 with isopropyl modified amidine groups (Figure 9A). For SPR results, the *K*_D_ value (12 nM) of DB2604 binding to target GAAAC sequence is slightly higher than the DB2528 *K*_D_ value (5 nM). To test the SPR binding results with these large complex compounds, the *K*_D_ of DB2604 was also evaluated by a fluorescence anisotropy titration of the compound with the GAAAC DNA sequence, and the *K*_D_ value is in good agreement with the results from SPR (*K*_D_ = 4 ± 2 nM).

An additional test of the binding selectivity of DB2528 was conducted with our competition mass spectroscopy method (Figure 10). In the first experiment binding of DB2528 to a mixture of DNAs are with no GC, A_4_T_4_; a DNA with a single GC, A_4_GT_4_; and a DNA with two G·C bps, GAAAC (left side of Figure 10). As can be seen in Figure 10, with these three DNAs, DB2528 binds only to GAAAC. In the second experiment DB2528 was mixed with a combination of two G·C bps containing sequences, GAAC, GAAAC, GAAAAC, and GAAAAAC. In another very exciting result, the compound only showed complex formation with the GAAAC DNA sequence, in excellent agreement with SPR results. This level of selectivity with such closely related DNAs is a strong validation of the module combination approach for recognition of more complex mixed sequences of DNA.

The results of this work, to link single G·C bp binding modules with a flexible linker for recognition of two G·C bps in the core sequence AGAAACT, has functional significance. With optimized design, synthesis and evaluation the linked heterocyclic-diamidine, DB2528 and several derivatives, have been obtained with an excellent match to the DNA minor groove shape and indexing of the compound with DNA functional groups at the floor of the groove. These compounds have syntheses that are reasonable in cost and time (Figure 9A) and provide a promising approach to develop a broad array of linked modular agents for competition with protein-DNA complexes and control of gene expression. These results illustrate what can be accomplished with modular compounds of this type in selective targeting of complex DNA. A number of additional compounds to recognize multiple G·C bps are being tested.

To obtain molecular insight of the very sequence-specific two G·C bps binding by the symmetrical dimer, DB2528, (Figure 9A) with the two G·C bps containing sequence, a 300 ns MD simulation of DB2528 with the GAAAC sequence (Figure 9B) was conducted as previously described [93,95]. The space-filling full view in Figure 11 at left side shows that DB2528 is able to match the curvature of the DNA minor groove and can cover a full turn of the double helix. The MD simulations showed the excellent contacts and van der Waals interactions that the compound makes with the minor groove molecular walls (Figure 11, left). A view of the complex structure at the upper part of the model (yellow circle) is shown in Figure 11 at the right side. This view shows a strong H-bond at a 2.02 Å distance between the *N* of *N*-MeBI and the dG-NH that points out into the groove. This interaction is a critical component of the G·C bp binding selectivity of the compound. There is also an H-bond between the amidine-NH that points to the floor of the groove and a dT=O (dT = O---H-N distance is 2.1 Å). The complex is additionally stabilized by water molecules that link DB2528 to the DNA binding site. The amidines form dynamic H-bonds to water molecules that enhance the stability of the H-bonds from the amidine and DNA (Am-N-H---O-H---N3-dA, distances are 2.1 Å and 2.0 Å respectively), right side of Figure 11.

At the other end of the complex (blue circle in Figure 11, left) the same type of H-bond and interaction pattern is obtained (not shown). The flexible -O-(CH_2_)_3_-O- linker is an appropriate length to effectively cover the AAA sequence between the two G·C bps. The compound excellent curvature, orientation, linker length and the specific interactions observed between DB2528 and the GAAAC sequence at the minor groove provide a rational explanation for the observed two G·C bps DNA recognition.

## 6. Conclusions

Description of the design, preparation and biophysical testing of entirely new types of heterocyclic amidine compounds with good solution and cell uptake properties that selectively recognize mixed A·T and G·C bp sequences of DNA is described in this report. These compounds are the first example of a series of rationally-designed agents for mixed sequence DNA recognition and they provide very different biotechnology reagents and new therapeutic candidates compared with those previously reported. Synthetic schemes for compounds that bind mixed base pair DNA sequences are given. To selectively recognize a G·C bp an H-bond acceptor must be incorporated into a compound. Pyridine, azabenzimidazole and thiophene-*N*-MeBI GC recognition groups have been appropriately incorporated in modules prepared by using both rational design and empirical optimization of the units. These modules can selectively and strongly recognize a single G·C bp in an AT sequence context. It is shown that in many cases relatively simple changes in substituents can convert a heterocyclic module from AT to GC recognition. Developments of lead compounds for recognition of G·C bps in DNA are presented. These compounds represent the first successful step for a new generation of diverse minor groove binders for mixed DNA sequence binding. Addition of more bulky groups in place of the *N*-methyl on the thiophene-*N*-MeBI GC recognition module shows that, like transcription factors, small molecules can be engineered for selective recognition of both sequence and shape in the minor groove. This dual recognition capability is a very useful step in our compound design efforts and provides compounds with much enhanced recognition selectivity. Modification of our initial sequence selective DNA binding agents has now moved to preparation and testing of larger compounds for recognition of more complex sequences of DNA with more than one G·C bps. Current research is focused on the preparation and linking of different modules to recognize these more complex mixed sequences of DNA.

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
