# Peer review of "A New Generation of Minor-Groove-Binding—Heterocyclic Diamidines That Recognize G·C Base Pairs in an AT Sequence Context"

_molecules, 2019, doi:10.3390/molecules24050946_

Round 1
Reviewer 1 Report
The manuscript ‘Design, Synthesis and Testing of a New Generation of Sequence-Selective, Minor-Groove-Binding Heterocyclic Diamidines’ by W. D. Wilson and coworkers, describes their own exciting advances on designing DNA minor groove binders based on diamidines stressing their ability of selectively recognize GC in the context of AT sequences. In my opinion these two aspects should be more highlighted in the title as well as in the abstract. Despite in the final sentences of the abstract the authors specific include the word ‘review’, in the middle part, one may even think that this is a scientific article instead of review. It would be important to define the aim and scope of this review and what it is different from the extensive work of Dervan. Indeed, the introduction should more explicitly mention Dervan hairpins due to unquestionable breakthrough for this topic. Also figure 1 should include the molecule BAPPA: N(1),N(3)-Bis(4-amidinophenyl)propane-1,3-diamine (Org. Lett.2010,12, 216) from Mascareñas. Also his work on bis-benzamidines and derivatives should be included in the introduction and not just be reduced two a couple of reference (12 and 13). Indeed, his work, where they also achieve the targeting of a G is even not include it (Chem. Sci, 2012,3, 2383).Such work is directly relevant for this review.In any case, this is an enjoyable piece of work, nicely written and comprehensive. Therefore, apart from this, and the lack of consistency in the size and style of figures/schemes, I see nothing else that needs to be changed. Nice work!
Author Response
Reviewer 1
The manuscript ‘Design, Synthesis and Testing of a New Generation of Sequence-Selective, Minor-Groove-Binding Heterocyclic Diamidines’ by W. D. Wilson and coworkers, describes their own exciting advances on designing DNA minor groove binders based on diamidines stressing their ability of selectively recognize GC in the context of AT sequences.
Comment:
In my opinion these two aspects should be more highlighted in the title as well as in the abstract. Despite in the final sentences of the abstract the authors specific include the word ‘review’, in the middle part, one may even think that this is a scientific article instead of review.
Answer:
The Reviewer has made an excellent point and in this revised manuscript we have rewritten the abstract and changed the Title of the article.
Comment:
It would be important to define the aim and scope of this review and what it is different from the extensive work of Dervan. Indeed, the introduction should more explicitly mention Dervan hairpins due to unquestionable breakthrough for this topic.
Answer:
We agree with the Reviewer and in this revised manuscript we have mentioned the scope and possibilities of synthetic diamidines. Also, in the introduction, we extensively discuss the Dervan and coworkers’ breakthrough on polyamides (Page 2 and 3, lines: 75-84). Also, we have incorporated some additional Dervan’s references, Refs 45-48.
Comment:
Also figure 1 should include the molecule BAPPA: N(1), N(3)-Bis(4-amidinophenyl)propane-1,3-diamine (Org. Lett.2010,12, 216) from Mascareñas.
Answer:
We again agree with the Reviewer that the compound and the reference should be included and apologize for this serious oversight. In this revised manuscript in Figure 1, we have added the structure BAPPA and a brief statement about BAPPA (page 2, lines: 67-71) with the reference (Ref. 40).
Comment:
Also his work on bis-benzamidines and derivatives should be included in the introduction and not just be reduced two a couple of reference (12 and 13). Indeed, his work, where they also achieve the targeting of a G is even not include it (Chem. Sci, 2012,3, 2383). Such work is directly relevant for this review.
Answer: We thank the Reviewer to point out this area that we have not discussed. In this revised manuscript we have described the important discoveries by Mascareñas and his coworkers (page 2, lines 67-74, and additional Refs. 40-42) on minor groove recognition.
Comment:
In any case, this is an enjoyable piece of work, nicely written and comprehensive. Therefore, apart from this, and the lack of consistency in the size and style of figures/schemes, I see nothing else that needs to be changed. Nice work!
Answer:
We thank Reviewer 1 for the valuable comments and suggestions which definitely improve the quality of this article. In this revised manuscript we are trying to improve consistency with all the figures and the schemes.
Reviewer 2 Report
Paul et. al. presented a report describing their latest works focused on the development of new compounds able to recognize mixed sequences of DNA. They started the review by discussing about the development of minor groove DNA binders and their application as antiparasitic agents. Then they moved towards the description of the design and optimization of molecules able to expand past pure AT sequences. They described how they approached this challenge by modifying AT selective heterocyclic cations by introducing H-bond acceptors to make the compounds able to selectively recognize a G·C base pair. Pyridine, azabenzimidazole, and thiophene-N-MeBI have been incorporated after opportune rational design and empirical optimization. In the last part of the review they chose to describe their last two published papers: Guo et. al. JACS 2018 and Guo et. al. Chem. Commun. 2017. They focused the last paragraphs on: i) the description of the affinity of compounds bearing a variety of substituents in place of N-methyl on the thiophene-N-MeBI GC recognition moiety and ii) on the synthesis of two units of N-methyl-benzimidazole-thiophene connected by a flexible linker that allows them to fit the shape and twist of the DNA minor groove while covering a full turn of the double helix.
Although this is a comprehensive review related to this topic a few questions have to be addressed before publication together with important English corrections.
- In the second paragraph while discussing about AT specific minor groove drugs as antiparasitic agents, this paper may deserve to be added to the references: Millan et. al. 8378–8391 Nucleic Acids Research, 2017, Vol. 45, No. 14.
- In pages 7 and 8 from line 257 and line 262 the authors discuss about the application of the DNA Shape algorithm of Rohs and coworkers. Following the references sited in this part of the review the reviewer could not find the paper where these data are published. Are these data only reported in the review or published somewhere else?
- The reviewer wants to point out a few typo mistakes present in the paper:
o Page 2 line 45: hindered should be replaced by hinders since it is an atemporal fact;
o Page 5 line 154: replace LiN (TMS)2 by LiN(TMS)2;
o Page 5 lines 173-174: scheme 2 between brackets at the end of the phrase is not really necessary considering that it is present at the beginning of the phrase;
o Page 6 lines 190 and 196: DB2277 shuold not be written in bold;
o Page 7 Figure 3: compound DB2429 is represented already in Figure 2. The reviewer understand that repeating the structure allows to compare better with its derivatives, but at this point in page 7 line 237 ”Figure 2” should be replace by Figure 2 and 4 or only 4;
o Page 9 line 301: In Figure 5 description “Representative SPR sensorgrams for A-C, DB2711 in the presence of AAATTT, AAAGTTT and AAAGCTTT hairpin DNAs. In A, C, the concentrations ...” should be replace by “Representative SPR sensorgrams for DB2711 in the presence of A) AAATTT, B) AAAGTTT, and C) AAAGCTTT hairpin DNAs. In A and C the concentrations…”. The modification makes the caption more readable;
o Page 10 line 305: “for global. kinetic” should be replaced by “for global kinetic”;
o Page 10 line 324: DB 2711 should be replaced by DB2711;
o Page 10 line 337: for the same reason mentioned above “Figure 6. (A, B, C) Torsional angle maps of a Ph-Am bond for DB2708, DB2759 and DB2762.” Should be replaced by “Figure 6. Torsional angle maps of a Ph-Am bond for A) DB2708, B) DB2759, and C) DB2762.”
o Page 11 line 354: “On-rates and off-rates and are plotted…” should be replaced by “On-rates and off-rates are plotted…”
o Page 11 line 354: “DB2528 (X)” it is not visible in Figure 7, it should be removed;
o Page 11 line 361: “…minor groove giving them…” them is not correct;
o Page 13 line 431: (Figure 9) should be replaced by (Figure 9B);
o Page 13 line 436: (Figure 9) should be replaced by (Figure 9A);
o Page 14 line 478: (Figure 9) should be replaced by (Figure 9A);
o Page 15 line 516: (Figure 9) should be replaced by (Figure 9A);
o Page 15 line 517: (Figure 9) should be replaced by (Figure 9B).
- At last, the reviewer wants to highlight a few phrase constructions that are very difficult to follow while reading the paper. The reviewer asks the authors to get through the review and rephrase them, and to look for all the others not listed in here, and check thoroughly the English.
o Page 2 lines 66-68: “The idea and results to redesign these successful diamidines to have significantly broadened sequence recognition capability and potentially expanded uses are reported in this review.”
o Page 3 lines 108-110: “The development took a major step forward with ideas from the Dickerson and Lown groups for modification of the minor groove binders, netropsin (Nt.) and distamycin (Dst.) [55–57].”
o Page 3 lines116-118: “Success in mixed sequence binding with both A·T and G·C bp recognition was achieved with polyamides after the finding by Wemmer and coworkers that it took two stacked or linked recognition units to make an effective minor groove polyamide for GC targeting [58,59].”
Author Response
Reviewer 2
Paul et. al. presented a report describing their latest works focused on the development of new compounds able to recognize mixed sequences of DNA. They started the review by discussing about the development of minor groove DNA binders and their application as antiparasitic agents. Then they moved towards the description of the design and optimization of molecules able to expand past pure AT sequences. They described how they approached this challenge by modifying AT selective heterocyclic cations by introducing H-bond acceptors to make the compounds able to selectively recognize a G·C base pair. Pyridine, azabenzimidazole, and thiophene-N-MeBI have been incorporated after opportune rational design and empirical optimization. In the last part of the review they chose to describe their last two published papers: Guo et. al. JACS 2018 and Guo et. al. Chem. Commun. 2017. They focused the last paragraphs on: i) the description of the affinity of compounds bearing a variety of substituents in place of N-methyl on the thiophene-N-MeBI GC recognition moiety and ii) on the synthesis of two units of N-methyl-benzimidazole-thiophene connected by a flexible linker that allows them to fit the shape and twist of the DNA minor groove while covering a full turn of the double helix.
Although this is a comprehensive review related to this topic a few questions have to be addressed before publication together with important English corrections.
Comment:
In the second paragraph while discussing about AT specific minor groove drugs as antiparasitic agents, this paper may deserve to be added to the references: Millan et. al. 8378–8391 Nucleic Acids Research, 2017, Vol. 45, No. 14.
Answer:
We agree with the Reviewer and in this revised manuscript we have added this reference (page 2, line 65, Ref. 39).
Comment:
In pages 7 and 8 from line 257 and line 262 the authors discuss about the application of the DNA Shape algorithm of Rohs and coworkers. Following the references sited in this part of the review the reviewer could not find the paper where these data are published. Are these data only reported in the review or published somewhere else?
Answer: Three mentioned DNA minor groove widths have been calculated by using Rohs’ online algorithm and the data are only published in this review. In this revised manuscript in Figure legend 4, we have also incorporated this statement.
The reviewer wants to point out a few typo mistakes present in the paper:
Comment:
Page 2 line 45: hindered should be replaced by hinders since it is an atemporal fact;
Answer:
We thank the Reviewer for finding this error, and it has been corrected in the revised version of this manuscript (Page 2, line 46).
Comment:
Page 5 line 154: replace LiN (TMS)2 by LiN(TMS)2;
Answer:
We again thank the Reviewer for finding this typo, and it has been corrected in the revised version of this manuscript (Page 5, line 170).
Comment:
Page 5 lines 173-174: scheme 2 between brackets at the end of the phrase is not really necessary considering that it is present at the beginning of the phrase;
Answer:
We agree with the Reviewer. We have deleted the brackets.
Comment:
Page 6 lines 190 and 196: DB2277 should not be written in bold;
Answer:
Again we agree with the Reviewer, and we make DB2277 un-bold (page 6, line 205 and 211)
Comment:
Page 7 Figure 3: compound DB2429 is represented already in Figure 2. The reviewer understand that repeating the structure allows to compare better with its derivatives, but at this point in page 7 line 237 ”Figure 2” should be replace by Figure 2 and 4 or only 4;
Answer:
We thank the Reviewer for finding this error, In this revised manuscript we have mentioned both Figures, Figure 2 and 3.
Comment:
Page 9 line 301: In Figure 5 description “Representative SPR sensorgrams for A-C, DB2711 in the presence of AAATTT, AAAGTTT and AAAGCTTT hairpin DNAs. In A, C, the concentrations ...” should be replace by “Representative SPR sensorgrams for DB2711 in the presence of A) AAATTT, B) AAAGTTT, and C) AAAGCTTT hairpin DNAs. In A and C the concentrations…”. The modification makes the caption more readable;
Answer:
We agree with the Reviewer and we have rewritten the figure legend 5 as following
“Representative SPR sensorgrams for DB2711 in the presence of A) AAATTT, B) AAAGTTT and C) AAAGCTTT hairpin DNAs. In A, C, the concentrations in the sensorgrams are 2-100 nM of each compound from bottom to top. In B, the concentrations of DB2711 from bottom to top are 15, 30, 40, 70, and 100 nM; In B, the solid black lines are best-fit values for global kinetic fitting of the results with a single site function.”
Comment:
Page 10 line 305: “for global. kinetic” should be replaced by “for global kinetic”;
Answer:
In this revised manuscript we have corrected this error.
Comment:
Page 10 line 324: DB 2711 should be replaced by DB2711;
Answer:
We thank the Reviewer for finding this typo, in this revised manuscript we have corrected this (page 10, line 339).
Comment:
Page 10 line 337: for the same reason mentioned above “Figure 6. (A, B, C) Torsional angle maps of a Ph-Am bond for DB2708, DB2759 and DB2762.” Should be replaced by “Figure 6. Torsional angle maps of a Ph-Am bond for A) DB2708, B) DB2759, and C) DB2762.”
Answer:
We agree with the Reviewer and we have rewritten the figure legend 6 as following
“Torsional angle maps of a Ph-Am bond for A) DB2708, B) DB2759 and C) DB2762. All calculations are performed at the B3LYP/6-31G* level of theory. The range of dihedral angle is from 0°-100°. The scanned dihedral is shown as the red bold line at 0̊.”
Comment:
Page 11 line 354: “On-rates and off-rates and are plotted…” should be replaced by “On-rates and off-rates are plotted…”
Answer:
In this revised manuscript the extra “and” has been deleted.
Comment:
Page 11 line 354: “DB2528 (X)” it is not visible in Figure 7, it should be removed;
Answer:
We agree with the Reviewer, In this revised manuscript we have incorporate DB2528(X) with a large and bold caption.
Comment:
Page 11 line 361: “…minor groove giving them…” them is not correct;
Answer: We again agree with the Reviewer. In this revised manuscript them is replaced by compounds (page 12, line 376).
Comment:
Page 13 line 431: (Figure 9) should be replaced by (Figure 9B);
Answer:
In this revised manuscript Figure 9 is replaced by Figure 9B.
Comment:
Page 13 line 436: (Figure 9) should be replaced by (Figure 9A);
Answer:
In this revised manuscript Figure 9 is replaced by Figure 9A.
Comment:
Page 14 line 478: (Figure 9) should be replaced by (Figure 9A);
Answer:
In this revised manuscript Figure 9 is replaced by Figure 9A.
Comment:
Page 15 line 516: (Figure 9) should be replaced by (Figure 9A);
Answer:
In this revised manuscript Figure 9 is replaced by Figure 9A.
Comment:
Page 15 line 517: (Figure 9) should be replaced by (Figure 9B).
Answer:
In this revised manuscript Figure 9 is replaced by Figure 9A.
Comment:
At last, the reviewer wants to highlight a few phrase constructions that are very difficult to follow while reading the paper. The reviewer asks the authors to get through the review and rephrase them, and to look for all the others not listed in here, and check thoroughly the English.
o Page 2 lines 66-68: “The idea and results to redesign these successful diamidines to have significantly broadened sequence recognition capability and potentially expanded uses are reported in this review.”
o Page 3 lines 108-110: “The development took a major step forward with ideas from the Dickerson and Lown groups for modification of the minor groove binders, netropsin (Nt.) and distamycin (Dst.) [55–57].”
· Page 3 lines116-118: “Success in mixed sequence binding with both A·T and G·C bp recognition was achieved with polyamides after the finding by Wemmer and coworkers that it took two stacked or linked recognition units to make an effective minor groove polyamide for GC targeting [58,59].”
Answer:
We thank the Reviewer for his careful reading and have modified all sentences of the Review that might be confusing (Page 3-4, lines 126-136).